# Current Challenges in the Treatment of the Omphalocele—Experience of a Tertiary Center from Romania

**DOI:** 10.3390/jcm11195711

**Published:** 2022-09-27

**Authors:** Elena Ţarcă, Elena Cojocaru, Laura Mihaela Trandafir, Alina Costina Luca, Răzvan Călin Tiutiucă, Lăcrămioara Ionela Butnariu, Claudia Florida Costea, Iulian Radu, Mihaela Moscalu, Viorel Ţarcă

**Affiliations:** 1Department of Surgery II-Pediatric Surgery, “Grigore T. Popa” University of Medicine and Pharmacy, 700115 Iaşi, Romania; 2Department of Morphofunctional Sciences I—Pathology, “Grigore T. Popa” University of Medicine and Pharmacy, 700115 Iaşi, Romania; 3Department of Mother and Child Medicine–Pediatrics, “Grigore T. Popa” University of Medicine and Pharmacy, 700115 Iaşi, Romania; 4Surgical Department, Iacob Czihac Military Emergency Clinical Hospital, 700483 Iaşi, Romania; 5Department of Medical Genetics, Faculty of Medicine, “Grigore T. Popa” University of Medicine and Pharmacy, 700115 Iaşi, Romania; 6Department of Surgery II-Ophthalmology, “Grigore T. Popa” University of Medicine and Pharmacy, 700115 Iaşi, Romania; 7Department of Surgery, Regional Institute of Oncology, I-st Surgical Oncology, "Grigore T. Popa" University of Medicine and Pharmacy, 700483 Iaşi, Romania; 8Department of Preventive Medicine and Interdisciplinarity, “Grigore T. Popa” University of Medicine and Pharmacy, 700115 Iaşi, Romania

**Keywords:** morbidity, infant mortality, risk factor, omphalocele, outcome

## Abstract

Omphalocele is a congenital abdominal wall defect with a constant incidence in recent decades, sometimes representing a real burden for neonatal intensive care units due to prolonged hospitalization and the evolution to death. In our study, we aimed to detect the main risk factors of an unfavorable evolution in the case of omphalocele. Methods: Retrospective cohort study of all neonates with omphalocele treated in our tertiary pediatric hospital during the last three decades; from 158 patients, 139 patients were eligible for the study. We tried to determine the risk of death using logistic regression model. Results: If the neonate develops sepsis, then there is an increased risk (13.03 times) of evolution to death. Similarly, the risk of death is 10.82 times higher in the case of developing acute renal failure, 6.28 times higher in the case of associated abnormalities, 5.54 in the case of developing hemorrhagic disease, and 3.78 in the case of conservative treatment (applied for giant omphalocele or severe chromosomal abnormalities). Prematurity increases by 3.62 times the risk of death. All six independent variables contributed 61.0% to the risk of death. The area under the ROC curve is 0.91, meaning that the diagnostic accuracy of our logistic regression model is very good for predicting the contribution of the six independent variables to the risk of death. Conclusion: Although in the past 30 years we witnessed several improvements in the antenatal diagnosis and management of omphalocele, survival rate remained constant, 47.5% overall. Much effort is still needed to eliminate the risk factors for death in this condition.

## 1. Introduction

Omphalocele and gastroschisis are the two most common congenital abdominal wall defects in newborns, and their incidence has a sinusoidal appearance, with periods of increase and decrease, probably depending on the variation of certain etiological factors [1]. Because of their lifelong impact on health and survival, congenital anomalies are increasingly recognized as a global health priority, abdominal wall defects being among the most challenging [2]. 

Although similar in terms of perinatal management and frequently analyzed together as "anterior abdominal wall defects", omphalocele and gastroschisis are two malformations with different etiologies and different antenatal and postnatal risk factors [3].

For omphalocele, the maternal risk factors are maternal age < 20 or >35 years, Afro-American ethnicity, maternal obesity, maternal disorders of glycemic control, and multiple births [2,4], while known patient risk factors associated with omphalocele are primarily chromosomal anomalies [5]. Omphalocele is especially common in patients with trisomy 18 (80–90% of cases) [6] and Beckwith–Wiedemann syndrome (10–66% of cases) [1]. Associated anomalies include cardiac (32%), chromosomal (17%), and central nervous system (8%) defects [7]. Genitourinary anomalies and diaphragmatic hernias are less commonly associated with omphalocele [8]. It is known that the association of genetic abnormalities and other birth defects are risk factors for an unfavorable evolution, and because in developed countries, the rate of antenatal diagnosis is around 96% on routine ultrasound, consequently, the rate of therapeutic abortion is high [9]. However, to what extent do these malformations increase the postnatal mortality rate? Further, if there are other predictors of death, these things are not yet established, and the treatment of omphalocele is not standardized either.

The aim of our study is to identify the main risk factors of an unfavorable evolution in the case of omphalocele in order to develop treatment strategies aimed at reducing short- and long-term health consequences. Active action focused on primary prevention and improving care is most effective when based on reliable information about the key indicators of the causes and outcomes of this malformation in the underlying population. 

## 2. Materials and Methods

We conducted a retrospective analytical study of neonates with omphalocele treated in our Pediatric Surgery Department. Inclusion criteria: all neonates treated for omphalocele from February 1991 to December 2021 in our tertiary care center. For the years 1991–2001, the data were collected from the patient files in the hospital archive; starting with the year 2002, the data were collected from the computer database of the hospital. The data were anonymized, and any possibility of direct or indirect identification of the patients’ identity was eliminated. Demographic data, antenatal diagnosis, gender, gestational age (GA), type of birth, birth weight (BW), associated abnormalities, time and mode of surgery, post-operative complications (PC), biological tests, presence of sepsis, length of hospitalization (LH), and mortality were analyzed. The terms used in the statistical analysis are defined below:-Acute renal failure is defined as a rapid fall in the rate of glomerular filtration, manifested clinically as a rapid increase in the serum levels of urea and creatinine above normal values for age, with an associated disruption of salt and water homeostasis [10];-Sepsis is defined as the presence of a positive blood culture or persistently abnormal clinical signs or inflammatory biomarkers and abnormal core temperature (>38.5 °C or <36 °C);-Associated abnormalities: two or more abnormalities in addition to omphalocele (central nervous system, cardiac, renal, skeletal or genital abnormalities);-Prematurity is considered as GA under 37 gestational weeks;-Hemorrhagic disease of the newborn is a life-threatening condition that is due to insufficient vitamin K levels in newborns as a result of various causes [11];-Cardiac abnormalities—we included here all the neonates with a minor or major cardiovascular malformation caused by abnormal cardiovascular development in the fetus.

The treatment method for newborns with omphalocele in our clinic is as follows: excision of the membranes and primary closure is attempted as a rule, based on the size of omphalocele and the ease of reduction of viscera under general anesthesia; arterial oxygen saturation and the need for increased ventilatory pressures are objectively assessed by the pediatric surgeon and the anesthesiologist to avoid abdominal compartment syndrome and acute renal failure. If excess abdominal tension with compromised ventilation or disproportional abdominal cavity is noted, a delayed closure with placement of a silo is performed. For giant omphalocele (more than 5–7 cm diameter with the liver outside the peritoneal cavity, depending also on the GA or when severe viscero-abdominal disproportion was noted) or when important chromosomal abnormalities are detected (trisomy 1, 13, 18), conservative treatment is applied. This means that once every two days, the newborn is taken to the operating room, the omphalocele membranes are washed with disinfectant solutions, and then, epithelializing ointments or epithelializing dressings are applied until the omphalocele turns into a ventral hernia (which will be treated surgically around the age of one year). In the last decade, with the appearance of high-performance incubators, most of these patients were treated conservatively in the neonatal intensive care unit. Approval for this retrospective study was granted by the Ethics Committee of “Saint Mary” Emergency Children’s Hospital.

### Statistical Analysis

We performed a descriptive statistical process separately over the three decades of the period 1991–2021 and analyzed a series of demographic and medical aspects within the studied group (demographic data, antenatal diagnosis, GA, type of birth, surgical approach, acute renal failure, associated abnormalities, BW, sepsis). In order to express the central tendency for the studied variables, which are not normally distributed, we used the median value. Because our data fail parametric assumption, to compare our indicators for the three decades, we applied the non-parametric Kruskal–Wallis test, which is a rank-based test. The results of this test located in the *p*-value column do not specify which specific groups for each variable are statistically significantly different from each other, but only that they are not the same across the three decades (*p* < 0.05). In order to highlight the significant differences between each pair of decades, we used post hoc tests, boldly marking specific groups of indicators that are statistically significantly different from each other. Then, we investigated whether there is an association between the following independent variables: prematurity, sex, surgical approach, acute renal failure, associated abnormalities, congenital heart disease, hemorrhagic disease, enterocolitis, blood transfusions, anemia, craniofacial dysmorphism and sepsis, and the risk of death of neonates with omphalocele. The chi-square test (χ^2^) and the contingency tables (cross tabulation) were used in the analysis, the studied variables being of nominal type. Then, the logistic regression was used to determine the risk of death of patients with omphalocele, a risk estimated based on independent variables previously validated by χ^2^. The nominal factorial variables underlying the proposed model were: sepsis (yes/no), acute renal failure (yes/no), associated abnormalities (yes/no), treatment (surgical/conservative), hemorrhagic disease (yes/no) and prematurity (yes/no). The death-dependent variable is a nominal dichotomous variable that can take the values yes/no. The regression equation used in the final model is: $$\frac{Prob\left(Death\right)}{Prob\left(Survive\right)}=e^{B_{0}}\times e^{B_{1}\left(Sepsis\right)}\times e^{B_{2}\left(Associated\;abnormalities\right)}\times e^{B_{3}\left(Acute\;renal\;failure\right)}\;\times e^{B_{4}\left(Treatment\right)}\times e^{B_{5}\left(Hemorrhagic\;disease\right)}\times e^{B_{6}\left(Prematurity\right)}$$

All *p*-values were two-sided, with values less than 0.05 considered statistically significant. Calculations were made using standard statistical package (JASP Team (2022). JASP (Version 0.16.1), University of Amsterdam, The Netherlands, https://jasp-stats.org/ accessed on 9 May 2022). 

For performing the regression analysis, using the stepwise method, we used an initial model that contains only the constant, and then, the computer searches for the predictor (out of the ones available) that best predicts the outcome variable (has the highest simple correlation with the dependent variable). If this predictor significantly improves the ability of the model to predict the outcome, then this predictor is retained in the model, and the computer searches for a second predictor (the variable that has the largest semi-partial correlation with the outcome) and so on. Each time a predictor is added to the equation, a removal test is made of the least useful predictor. If the first version of our model contains only the first predictor (sepsis), which has the highest simple correlation with the dependent variable, by using the stepwise method, the final model contains six independent variables, valid for the dependent dichotomic variable death prediction: sepsis, acute renal failure, associated abnormalities, conservative treatment, hemorrhagic disease, and prematurity.

An often-used measure of goodness-of-fit for evaluating the fit of a logistic regression model measures simultaneously is the sensitivity (true positive) and the specificity (true negative) for all possible cutoff points. Sensitivity and specificity pairs for each possible cutoff point and plot sensitivity on the y-axis by (1-specificity) on the x-axis using the ROC (Receiver Operating Characteristic) curve are calculated. The area under the ROC curve ranges from 0.5 and 1.0, with larger values indicative of better fit.

One of the assumptions in logistic regression is explanatory variables should not be highly correlated with each other. Multicollinearity can cause unstable estimates and inaccurate variances, which affects confidence intervals and hypothesis tests. Testing multicollinearity is important for checking correlations between independent variables of the regression model. Tolerance is a useful indicator in testing the multicollinearity of independent variables. Multicollinearity is indicated by tolerance values close to 0.

## 3. Results

From a total of 158 patients, 139 newborns with omphalocele were eligible for this study, being treated in our hospital during the analyzed period. Nineteen patients with incomplete medical records or operated in another clinic were excluded from the analysis. There was an incidence of 46/53/40 newborns per decade. The male/female ratio was 1.17 per total. The mean rate of associated abnormalities (more than two) per total was 34.5% and was constantly decreasing per decade, from 43.4% to 33.9% and 25%, probably as an effect of increased antenatal diagnosis and therapeutic abortion of multiple malformations. A rate of 16.5% out of the total had genetic abnormalities (six patients with Beckwith–Wiedemann syndrome, one patient with Fryns syndrome, six patients with Down syndrome, six with trisomy 18, three with trisomy 13, and one patient with trisomy 1). Out of 59 patients diagnosed with cardiac anomalies, 29 had major cardiac malformations (20.8% of all patients) (ventricular septal defect, severe pulmonary valvular stenosis, tetralogy of Fallot, transposition of the great arteries, hypoplastic left heart syndrome, atrioventricular canal).

In almost 60% of patients, it was possible to integrate the viscera in the abdomen and close the abdominal wall per primam or with the aid of a silo bag; the rest of the patients were treated by conservative methods.

Over the analyzed period, the median age of the mothers was 27 years, with a significant increase in the last decade to a median of 31 years. The rate of antenatal diagnosis and cesarean births increased significantly, and the gestational age decreased, as can be seen in Table 1. The survival rate was constant, around 47.5%; although the survival rate in the last decade was 55%, the increase was not statistically significant. The evolution of other parameters (BW, time to surgery, age at death, LH) can also be observed in Table 1. Additional demographic and personal characteristics of the patients are presented per decade and per total analyzed period in Appendix A. 

In order to highlight the significant differences between each pair of decades, we used post hoc tests, boldly marking specific groups of indicators that are statistically significantly different from each other. Thus, for mothers age, the median age of the third decade was significantly higher compared to the second decade (31 > 25). 

To determine if a relationship exists between several independent categorical variables (prematurity, sex, treatment, sepsis, acute renal failure, associated abnormalities, cardiac abnormalities, hemorrhagic disease, enterocolitis, blood transfusions, anemia, and craniofacial dysmorphism) and the risk of death, we used Pearson’s χ^2^ test. As can be seen from the analysis in Table 2, there is a significant association between some independent variables and the risk of death (*p* < 0.05). Thus, in the case of conservative treatment compared to surgery, the chances of death are 2.09 times higher. In the case of sepsis, the patient’s chance of dying is 7.87 times higher. Acute renal failure also increases the chance of death by 18.78 times. The presence of associated abnormalities has a negative impact on the patient’s chances of survival, increasing the risk of death by 3.75 times. Hemorrhagic disease increased the risk of death 10.94 times, blood transfusions 2.37 times, prematurity 3.29, and craniofacial dysmorphism 2.06 times.

Logistic regression was performed to ascertain the effects of prematurity, sepsis, acute renal failure, associated abnormalities, conservative treatment, hemorrhagic disease, blood transfusions, and craniofacial dysmorphism on the risk of death. The step 6 model presented in our study is the best, having a value of 0.61 for Nagelkerke’s R-square, which proves that all six independent variables in our model contribute 61.0% to the risk of death (Appendix A).

According to model summary, we observe that values of “–2 Log likelihood” are decreasing from one step to another, from 165.685 in step 1 to 108.090 in step 6. This shows us the fact that our 6th model (the final model with six predictors) fits the data better than the previous five models. The confusion matrix (Appendix A) shows that the 56 true-negative and 62 true-positive cases were predicted by the model, while the errors, false negatives, and positives were found in 11 and 10 cases, respectively. This is confirmed in the performance metrics where sensitivity (% of cases that had the outcome correctly predicted) is 84.93% and specificity (% of cases predicted as not having the outcome) is 84.85%. In the regression model, the six independent variables considered (sepsis, acute renal failure, associated abnormalities, conservative treatment, hemorrhagic disease, and prematurity) can predict which value of the dependent one (death) is observed in the dataset 84.89% of the time.

In the final table, model coefficients (Table 3), it can be seen that each independent variable considered in the model has a significant impact on the predicted variable (*p* < 0.05). 

Therefore, we shall have the logistic regression equation:$$\frac{Prob\left(Death\right)}{Prob\left(Survive\right)}=\left(0.06\right)\times \left(13.03\right)\mathrm{Sepsis}\times \left(6.28\right)\mathrm{Associated}\;\mathrm{abnormalities}\;\times \left(10.82\right)\mathrm{Acute}\;\mathrm{renal}\;\mathrm{failure}\;\times \left(3.78\right)\mathrm{Treatment}\times \left(5.54\right)\mathrm{Hemorrhagic}\;\mathrm{disease}\times \left(3.62\right)\mathrm{Prematurity}$$

If the sepsis is present, then there is an increased risk (13.03 times) of evolution to death. Similarly, as seen from the logistic regression equation, the risk of death is 10.82 times higher in the case of developing acute renal failure, 6.28 times in the case of associated abnormalities, 3.78 in the case of applying a conservative treatment, 5.54 in the case of hemorrhagic disease, and 3.62 in the case of prematurity. All six independent variables contribute 61.0% to the risk of death.

The area under the ROC (receiver operating characteristic) curve (AUC) is an aggregated metric that evaluates how well the logistic regression model classifies positive and negative outcomes at all possible cutoff points. The area under the curve (Figure 1) is 0.91. In addition, AUC is significantly different from 0.5 since the *p*-value is 0.000, meaning that the diagnostic accuracy of our previously presented logistic regression model is very good for predicting the contribution of the six independent variables (sepsis, acute renal failure, associated abnormalities, conservative treatment, hemorrhagic disease, and prematurity) to the risk of death in the case of omphalocele.

Multicollinearity is indicated by tolerance values close to 0, but since we do not have this situation for the variables of the model presented (all tolerance values are between 0.783 for sepsis and 0.956 for hemorrhagic disease), we can state that our predictors are not too highly correlated to each other. In the case of VIF (variance inflation factor), values exceeding 10 are often regarded as indicating multicollinearity, but in weaker models, which is often the case in logistic regression, values above 2.5 may be a cause for concern. In the model proposed by us, the VIF values are between 1.046 in the case of hemorrhagic disease and 1.278 for the sepsis. Simple arithmetic means of the six VIF’s calculated for our model is equal to 1.144, a value that is very close to 1 and once again confirms that collinearity is not a problem for the model (Appendix A).

## 4. Discussion

Congenital anomalies are among the most prevalent cause of infant deaths in many countries [12]. Although the main risk factors for omphalocele appearance have been extensively studied and determined, just a few independent risk factors for mortality among omphalocele newborns have been reported previously in the literature; efforts have been made in the last years to detect those determinants of increased morbidity and postoperative mortality in order to eliminate or improve them. We conducted a study that is unique in our country due to the large group of patients and due to the fact that it covers a long period of time (30 years), and we found that associated abnormalities, sepsis, acute renal failure, conservative treatment methods, hemorrhagic disease, and prematurity are the main risk factors associated with death in omphalocele patients in our country.

Young maternal age or advanced maternal age and multiparity are considered important risk factors in maternal–fetus–neonatal health and are associated with higher fetus–neonatal morbidity and mortality [13]. Advanced maternal age is increasing over the last two decades (between 5% and 7% among mothers of babies affected with congenital abnormalities), and is affecting the prevalence of aneuploidy [14,15]. In agreement with these prior studies, we observed that over the analyzed period, the median age of the mothers with an omphalocele baby was 27 years, with a significant increase in the last decade to a median of 31 years. Omphalocele seems to occur more frequently in males than females [5], which was also confirmed by our study, with a rate of 1.17.

Routine prenatal screening and diagnosis of abdominal wall defects and of associated anomalies is considered standard of care, and nowadays, over 90% of cases are diagnosed prenatally in developed countries [1]. In cases of liver herniation, prenatal ultrasound can detect an omphalocele at 9–10 gestational weeks. Prenatal diagnosis in cases without liver herniation can be made reliably between 12 and 16 gestational weeks [8]. In the Netherlands and Sweden, an improved prenatal detection rate together with the high rates of associated anomalies in omphalocele increased the pregnancy termination rate for abdominal wall defects up to 60% [9,16]. A similar effect may apply to Romania in the future to all congenital anomalies that can be detected prenatally, and the prevalence of congenital malformations may decrease; it is known that low socioeconomic status is associated with health providers with lower capacity for prenatal diagnosis and therefore with lower access to termination of pregnancy due to fetal anomalies [12]. The rate of antenatal diagnosis increased significantly in our country in the last decades (35%), but it has not yet reached the level of developed countries. After the collapse of the communist regime in Romania in 1989, our Ministry of Health adopted a standard of prenatal care practice for all women. The package of free services included risk assessments, medical examinations, laboratory tests and screenings, behavioral screenings, and prenatal counselling sessions [17,18]. However, after 10 years of implementation, 9 out of 10 pregnant women did not receive all the necessary medical services; in the year 2010, seventy-eight percent of mothers underutilized prenatal care services, and after almost 30 years, the antenatal diagnosis rate of congenital malformations detectable antenatally is only 30–50% [3,18]. Young maternal age, marital status—unmarried, low educational level, rural environment, alcohol and tobacco consumption, and Roma ethnicity are the factors most frequently associated with the low use of the free prenatal medical services offered by the health services in our country [18,19]. In a recent study, although the rate of therapeutic abortion was 59% for omphalocele, the rate of abnormalities associated with newborns was 62% (including minor abnormalities) [9]. The survival rate in this cohort was still high, and this can be explained by a positive selection of cases and due to improvements in neonatal intensive care during the years. In another study, after excluding all the fetuses with abnormal karyotype or other malformations, the survival rate at birth was 68% among the 79 isolated fetal omphaloceles [20]. In the last decade, we registered a 55% overall survival rate. Thirty-four percent of our omphalocele cases had two or more associated abnormalities, and 16.5% had chromosomal or other genetic abnormalities, which agrees with other studies reporting rates of associated abnormalities between 31% and 50% and rates of genetic abnormalities between 8% and 32% [4,21,22,23]. It is well-known from the specialized literature that genetic anomalies (such as trisomy 13, 18, or 1), major cardiac malformations, and pulmonary hypoplasia associated with large omphalocele (conservative treatment) are risk factors for increased morbidity and mortality in case of omphalocele. We also demonstrated in our study that two or more associated abnormalities increase the mortality risk by 3.75 times (Table 2) and 6.28 times if the logistic regression is applied, while cardiac abnormalities increase this risk 1.61 times (Table 2) and craniofacial dysmorphism 2.05 times (Table 2). If all these serious anomalies were detected antenatally, and therapeutic abortion would be recommended, these newborns would no longer die postnatally, increasing neonatal mortality in our country [19]. 

It is also known that associated congenital malformations and chromosomal abnormalities are risk factors for prematurity [24,25], so the lack of antenatal diagnosis and therapeutic abortion for these serious abnormalities led to an increase in the frequency of premature births in our group. We demonstrated that prematurity increases the risk of death by 3.29 times in the case of patients with omphalocele (by 3.62 times in the regression model). 

After in utero diagnosis of an abdominal wall defect, the care of the pregnant mother should be transferred to a tertiary care center for further counseling and treatment by an interdisciplinary team encompassing obstetricians, neonatologists, and pediatric surgeons. Although cesarean delivery has been shown to be beneficial for the fetus only in the case of giant omphaloceles with an increased risk of dystocia or rupture of the membranes, the rate of cesarean births has increased significantly in our country with increasing rates of antenatal diagnosis [2]. After birth, the patient with omphalocele is immediately taken into the neonatal intensive care unit, is treated for electrolyte or thermal imbalances, is investigated for the presence of associated abnormalities, and the therapeutic management is established: surgery or conservative treatment depending on the size of the omphalocele and associated findings. Because omphalocele is not a surgical emergency, and numerous studies report the importance of detecting associated abnormalities [6,21], in the last decade, we postponed the operative time in our clinic (from 18 h to 40 h on average) in favor of performing more preoperative investigations and treating associated imbalances; this fact led to a slight increase in survival in recent years. Special attention must be paid to premature newborns who associate hydro electrolytic, acid–base imbalances or vitamin K deficiency; we demonstrated in our study that hemorrhagic disease increases the risk of death by 10.93 times (5.54 times within the logistic regression model).

For omphalocele, an overall mortality rate of 14–30% has been reported in developed countries and between 41 and 50% in developing ones [4,21,26]. In a multi-country study reporting an overall mortality rate of 32.1%, one-third of children with omphalocele died before age 5 years, and most deaths occurred during the first 24 h after birth, followed by the first week of life [21]. This is confirmed by our study, which found a median age at death of 7 days. In different studies, adverse outcomes were associated with some neonatal variables such as lower Apgar score, initial mechanical ventilation, late-onset sepsis, pulmonary hypoplasia, and low lung volumes by fetal MRI [27,28]. For genetic abnormalities alone, another study found that live-born children with omphalocele and associated chromosomal anomaly had a tenfold higher risk of mortality than those with isolated omphalocele [9]. Regarding the approach to omphalocele closure, a recent study of the giant omphalocele population demonstrated an unfavorable outcome related to the repair other than primary [29], as we also demonstrated in the current study. The conservative treatment is a very good option in the case of giant omphalocele, avoiding the abdominal compartment syndrome [7]. The increased risk of mortality associated with this type of treatment is not necessarily related to the method of treatment itself but to the prolonged period of hospitalization (with the risk of nosocomial infections) and pulmonary hypoplasia associated with giant omphalocele [2]. In order to reduce the period of hospitalization and the related infectious risks as much as possible, we have recently tried to close the abdominal wall defect sometimes using a low-pressure negative-pressure system. We have reported these patients as being treated surgically, which is why the number of patients treated conservatively in the last decade is lower (45% versus 30%). Due to this aspect as well as other positive factors, we managed to increase the survival rate from 41.5 to 55% in the last decade. The mortality rate for omphalocele in our clinic is still very high per total period, but this is more due to the lack of antenatal diagnosis and lack of therapeutic abortion in case of detection of serious associated abnormalities (such as the genetic ones) and less due to the surgical/conservative management itself.

Our study highlighted that if sepsis is present, then there is an increased risk (7.87 times) of evolution to death (13.03 according to the final regression model). A 2018 review of predictors of mortality in neonates with giant omphalocele found that sepsis was the leading cause of death: in 56.6% of cases, respectively [28]. Antenatal diagnosis, rebalancing of biological parameters and the prophylaxis of hemorrhagic disease of the newborn, bringing the newborn as soon as possible to the neonatal intensive care unit, performing surgery in the best aseptic conditions, avoiding the abdominal compartment syndrome, and early antibiotic therapy are basic conditions to prevent wound infection, sepsis, or other complications. Similarly, the risk of death is 10.82 times higher in the case of developing acute renal failure, 5.54 times higher in the case of hemorrhagic disease, 6.28 times in the case of associated abnormalities, and 3.78 in the case of applying a conservative treatment. All six independent variables contributed 61.0% to the risk of death. According to the ROC curve, the diagnostic accuracy of the presented logistic regression model in our study was very good for predicting the contribution of the six independent variables to the risk of death; also, multicollinearity was not a problem for our model. 

Therefore, in the case of congenital anomalies in general and omphalocele in particular, both risk factors for the occurrence of this malformation and risk factors for an unfavorable postnatal evolution have been identified. Education combined with preconception and premarital counseling are important prevention strategies, focusing on increasing awareness to allow couples to make more informed choices [14,30,31]. Health education programs should be implemented in schools to emphasize the importance of preventing unwanted pregnancies at a young age and of a planned pregnancy at the optimal age (20 to 30 years), to ensure adequate periconceptional folic acid intake and avoid the use of toxic substances [2,3,14,30]. In terms of perinatal management, increasing the rate of antenatal diagnosis of severe associated malformations or giant omphalocele with abdominal-visceral disproportion will lead to therapeutic abortion and decreased postnatal mortality. Proper surgical treatment, careful monitoring of abdominal pressure, avoiding the use of empirical antibiotic therapy, and ensuring the strictest aseptic measures in neonatal intensive care units will prevent acute renal failure and sepsis and decrease morbidity and mortality in the case of omphalocele [2,3,32].

Finally, our study draws attention to the usefulness of a congenital anomalies surveillance system in every country as a source of information to identify those neonates at risk and guide prevention actions.

The strength of our study lies in the large cohort of approximately one-fifth of the Romanian patients. This is the largest number of patients that has been analyzed in a single study in Romania; the generalization of the results to the entire population of the country will, however, be done with caution. Another positive aspect of the study was the accessibility of individual level information on patient characteristics, clinical presentation of the defect, sociodemographic factors, and comorbidities and being a single center study in addition to the uniformity of the applied therapeutic management. Furthermore, we had information on the exact cause of death, on risk factors for a poor prognosis, and on time or type of surgery and postoperative complications. Our study provides valuable information for neonatologists, pediatric surgeons, and public health professionals in Romania and around the world in planning and providing maternal-fetal services. It also provides new data for use in future comparisons in the analysis of morbidity and mortality linked to omphalocele.

Analysis of the medical care of patients with abdominal wall defects in our country and comparing the results to the published literature is somewhat challenging since no nationwide clinical registry exists and epidemiologic studies are difficult to perform. Therefore, there is a limitation intrinsic to this study, as the review was retrospectively performed, and it is a single-center study. Furthermore, our study covers a period of transition between the almost total absence of antenatal diagnosis (with newborns with multiple associated anomalies) and the period when antenatal diagnosis began to lead to the recommendation of therapeutic abortion and the birth of patients with fewer associated anomalies. As we have previously shown, the accessibility of pregnant women to the free medical services offered by the Romanian state has increased progressively in the last 30 years, but it is still far below the level of developed countries. These facts led to the appearance of a non-homogeneous study group, and more biases can result.

## 5. Conclusions

In the last three decades, we have witnessed improvements in antenatal diagnosis and management of congenital malformations due to enhanced integration between the disciplines of maternal fetal medicine, neonatology, and pediatric surgery. Increasing the rate of antenatal diagnosis of severe associated malformations or giant omphalocele with abdominal-visceral disproportion will lead to therapeutic abortion and decreased postnatal mortality. In addition, some factors such as sepsis, acute renal failure, hemorrhagic disease, and treatment methods may still be ameliorated for better outcomes of these patients.

## Figures and Tables

**Figure 1 jcm-11-05711-f001:**
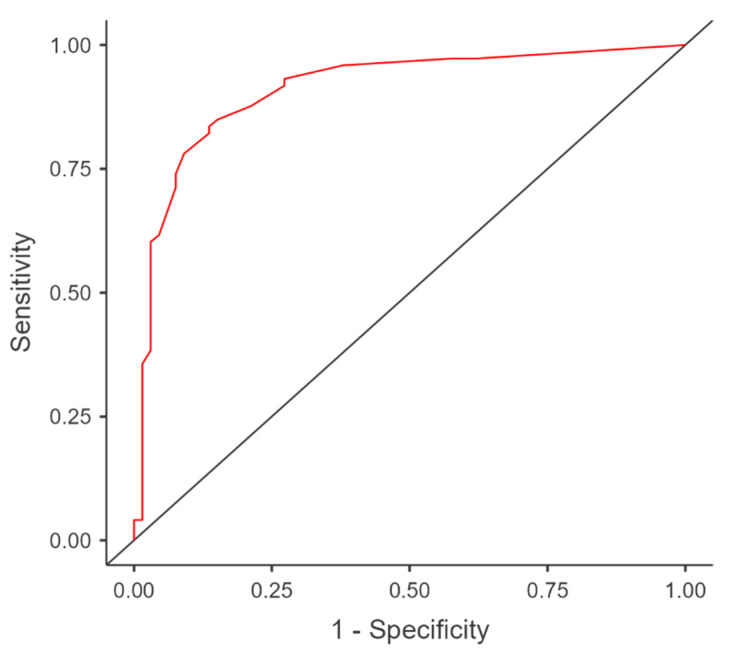
ROC curve for the logistic regression model (AUC = 0.91). ROC, receiver operating characteristic; AUC, area under the ROC curve.

**Table 1 jcm-11-05711-t001:** Descriptive statistics of the analyzed patients.

		1991–2000	2001–2010	2011–2021	Overall	*p*-Value *
Frequency of occurrence		46	53	40	139	
Mothers age (between 15 and 45 years old)	Median Value	26	**25**	**31**	27	**0.02**
Gestational age (weeks)		**39**	**38**	38	38	**0.006**
Birth weight (between 900 and 4700 grams)		2750	2700	2900	2750	0.576
The age at the time of surgery (hours)		18	12	40	20.5	0.149
Age at death (days)		5.4	7	10	6.9	0.174
Length of hospitalization for survivals (days)		10	14	12.5	12	0.687
The survival rate	Percentage	47.83	41.51	55	47.48	0.437
The rate of antenatal diagnosis		**6.52**	**15.09**	**35**	17.99	**0.002**
The rate of cesarean births		**19.57**	33.96	**47.5**	33.09	**0.023**
The rate of conservative treatment		45.7	45.3	30	41	0.247
The rate of sepsis		34.78	41.51	20	33.09	0.09
The rate of acute renal failure		13.04	24.53	25	20.86	0.282
The rate of cardiac abnormalities		**30.44**	**28.3**	**75**	42.45	**<0.001**
The rate of craniofacial dysmorphism		32.61	20.76	25	25.9	**0.401**

* Non-parametric Kruskal–Wallis Test. The statistically significant differences were bolded.

**Table 2 jcm-11-05711-t002:** Chi-square tests for statistical variables considered individually and the risk of death.

Risk of Death	χ^2^ Value	df	*p*	OR	95% CI	
**Prematurity**	8.993	1	0.003	3.295	1.482	7.329
Sex	0.224	1	0.636	1.175	0.602	2.293
Sepsis	24.964	1	0.000	7.871	3.297	18.792
Acute renal failure	24.206	1	0.000	18.783	4.252	82.960
Associated abnormalities (two or more)	12.234	1	0.000	3.755	1.755	8.040
Conservative Treatment	4.386	1	0.036	2.085	1.044	4.160
Cardiac abnormalities	1.903	1	0.168	1.612	0.817	3.182
Hemorrhagic disease	19.008	1	0.000	10.937	3.118	38.369
Enterocolitis	2.232	1	0.135	2.167	0.772	6.077
Blood transfusions	6.098	1	0.014	2.369	1.188	4.724
Anemia	0.207	1	0.649	1.168	0.599	2.276
Craniofacial dysmorphism	4.272	1	0.040	2.056	1.034	4.087

0 cells (0.0%) have expected count less than 5. OR, odds ratio; CI, confidence Interval.

**Table 3 jcm-11-05711-t003:** Model coefficients.

Parameter	Estimate	Standard Error	z	Wald Statistic	df	*p*	OR
Sepsis (yes)	2.567	0.604	4.249	18.053	1	<0.001	13.031
Associated abnormalities (yes)	1.837	0.566	3.245	10.532	1	0.001	6.276
Acute renal failure (yes)	2.382	0.885	2.691	7.243	1	0.007	10.824
Conservative Treatment (yes)	1.330	0.509	2.610	6.814	1	0.009	3.780
Hemorrhagic disease (yes)	1.712	0.753	2.273	5.166	1	0.023	5.543
Prematurity (yes)	1.286	0.557	2.307	5.323	1	0.021	3.618
Intercept (constant)	−2.858	0.558	−5.118	26.192	1	<0.001	0.057

Note. Death level “YES” coded as class 1; OR, odds ratio (e^B^), indicating the effect of the predictor variable on the predicted variable for the logistic regression analysis.

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
