# Peer review of "Current Challenges in the Treatment of the Omphalocele—Experience of a Tertiary Center from Romania"

_jcm, 2022, doi:10.3390/jcm11195711_

Round 1
Reviewer 1 Report
I suggest correcting the title in "Current challenges in the treatment of omphalocele - experience of a tertiary center from Romania"
You state that, due to the type of study, the collection of informed consents was waived. I assume that the ethics committee of your institution agreed with that.
Please provide the ethics approval number.
Did you have all the data in a computer database since 1991, or did you review some of it through a paper database?
The section from lines 147 to 156 falls under the domain of the statistical analysis section. Please correct the section. Below the table, write what the bold values mean.
How do you explain the increase in age at the time of surgery, from 18 and 12 to 40 hours?
Since this is not a large number of patients, it would be very useful to add tables with complete data from your study as supplementary material.
Could you also add the odds ratio values in percentages in the text (in brackets).
Also, please move part of the text from lines 169 to 184 to the materials and methods section. In the results section, please write only the parts related to the results.
The explanation of Table 3 might be unclear for the readers. Please write the section under table 3 more clearly (note).
For all abbreviations that appear in the tables, please state the full name in the note.
From lines 209 to 212, you repeat the previously stated results. That is unnecessary.
Author Response
Dear Reviewer, thank you for your time evaluating our manuscript. Your recommendations and comments helped us greatly improve our manuscript.
- I suggest correcting the title in "Current challenges in the treatment of omphalocele - experience of a tertiary center from Romania"
Response: We corrected the title according to your suggestion.
- You state that, due to the type of study, the collection of informed consents was waived. I assume that the ethics committee of your institution agreed with that. Please provide the ethics approval number. Did you have all the data in a computer database since 1991, or did you review some of it through a paper database?
Response: The collection of patient data was performed retrospectively; for the years 1991 - 2001, the data were collected from the patient files in the hospital archive; starting with the year 2002, the data were collected from the computerized data network of the hospital. The data were anonymized and any possibility of direct or indirect identification of the patients' identity was eliminated. We added these specifications in the manuscript (line 78-82). The approval of the ethics commission was obtained for the conduct of the study. We have attached the registration number.
- The section from lines 147 to 156 falls under the domain of the statistical analysis section. Please correct the section. Below the table, write what the bold values mean.
Response: We corrected the statistical analysis section accordingly. The values ​​between which statistically significant differences were detected were bolded. We added the explication below the table.
- How do you explain the increase in age at the time of surgery, from 18 and 12 to 40 hours?
Response: Because omphalocele is not a surgical emergency and numerous studies report the importance of detecting associated abnormalities, in the last decade in our clinic we postponed the operative time (from 18 hours to 40 hours on average) in favor of performing more preoperative investigations and treating associated imbalances; this fact also led to a slight increase in survival in recent years. We added this phrase in our manuscript (line 364-369).
- Since this is not a large number of patients, it would be very useful to add tables with complete data from your study as supplementary material.
Response: We added supplementary data in the main text (Table 1) as well as supplementary materials (Table S1).
- Could you also add the odds ratio values in percentages in the text (in brackets).
Response: Since odds ratios are numbers greater than 1, we chose this way of expression in the manuscript, for an easier reading of the text.
- Also, please move part of the text from lines 169 to 184 to the materials and methods section. In the results section, please write only the parts related to the results.
Response: We proceeded according to your recommendations
- The explanation of Table 3 might be unclear for the readers. Please write the section under table 3 more clearly (note).
Response: At the suggestion of the other reviewer, we moved table 3 to Supplementary materials (Table S2), and we moved the explanations to the Statistics chapter; thank you for your suggestion.
- For all abbreviations that appear in the tables, please state the full name in the note.
Response: We proceeded according to your recommendations.
- From lines 209 to 212, you repeat the previously stated results. That is unnecessary.
Response: We corrected.
Thank you again for reviewing our manuscript,
Elena Țarcă, MD, PhD
Author Response
Dear Reviewer, thank you for your time evaluating our manuscript. Your recommendations and comments helped us greatly improve our manuscript.
- The authors should be commended on following and reporting outcomes from a large cohort over many years. However, I have several serious concerns about the manuscript, and find that it leaves me with more questions than answers. Among these are the fairly large discrepancies between reported patient population and outcomes and those reported in the existing literature, such as with the incidence of associated abnormalities (34.5% in the present study, compared to up to 80% in the published literature (Benjamin et al., J Ped Surg 2014)), and survival of 47.5% (compared to ~68% overall (Nembhard et al., Birth Defects Res. 2020), and ~90% in the absence of associated anomalies (Tassin et al., Prenat Diagn 2013). The authors do address these discrepancies briefly in the discussion, but I would like to know more about why their study population is unique.
Response: The uniqueness of our study consists in the fact that it covers a period of transition between the almost total absence of antenatal diagnosis (with newborns with multiple associated anomalies) and the period when antenatal diagnosis began to lead to the recommendation of therapeutic abortion and the birth of patients with fewer associated anomalies. In addition, to statistically analyze the correlation between the presence of associated anomalies and death, we considered those patients who had at least two associated anomalies in addition to omphalocele (central nervous system, cardiac, renal, skeletal or genital abnormalities), as mentioned in the Methods chapter – line 95-96. In the main text and table 1, we added the rate of cardiac abnormalities, craniofacial dysmorphism and genetic abnormalities. We also added supplementary materials with more data on our studied population.
Regarding the survival rate, in the last decade, the survival rate of our patients was 55%, close to the one cited in current studies (Nembhard et al., Birth Defects Res. 2020). You also referred to Tassin's study (Tassin et al., Prenat Diagn 2013), reporting a 90% survival rate. It is true, but they excluded all the fetuses with abnormal karyotype or other malformations and the survival rate at birth was 68%. Only among the latter, the survival rate after the neonatal period was 90%. (“Among 153 fetal omphaloceles diagnosed before 14 WG, 74 were excluded because of abnormal karyotype or other malformations. Among the 79 isolated fetal omphaloceles, the survival rate at birth was 68% (54/79), with a global morbidity rate of 33% (18/54). Of the live born fetuses, 92.6% (50/54) survived the neonatal period.” - Tassin et al., Prenat Diagn 2013). We added this comment in our manuscript (line 335-3151).
- Perhaps more importantly, there are large holes in the data presented that would be highly relevant to such a report. For example, the authors note that associated anomalies, sepsis, acute renal failure, and conservative management were predictive of mortality, but do not state the incidence of any of these predictors – for example, how many patients developed sepsis?
Response: From the total of 139 cases studied, 46 patients (33.09%) developed sepsis. From the total of 139 cases studied, 29 patients (20.86%) developed acute renal failure. Out of the total of 139 cases studied by us, 48 patients (34.5%) had two or more associated abnormalities. From the total of 139 cases studied by us, 57 patients (41.0%) had conservative treatment. We added these data in our manuscript (line 164-170 and Table 1).
- Also, what other variables, if any, were analyzed in their statistical models?
Response: We mentioned in table 2 (which was redone) the other studied indicators and commented in the text (line 204-206).
- How long were patients followed in this study? These are critical pieces of information for the reader. The authors clearly have a large patient series that is highly representative of omphalocele care in Romania. With further data collection, reporting, and analysis, they may have substrate for a compelling manuscript. Further comments are outlined below.
Response: Our current study focused on the analysis of the neonatal period and the data were taken from the first admission of the patient. For most of the patients, we have the data from subsequent admissions and we have followed their evolution over time; for the patients lost from the records, we intend to carry out another study in which we will look for them and find out their evolution, in order to analyze and publish the results.
Major Critiques
- More information is needed on patient characteristics and demographics. These should include: Breakdown of associated anomalies (including congenital heart disease, pulmonary hypertension/hypoplasia, non-chromosomal genetic syndromes such as BeckwithWiedemann). Many of these, heart disease in particular, are recognized as highly predictive of outcome but were not listed among predictors in the regression model.
Response: We added the requested information in text (line 164-170), in Table 1 and in the supplementary materials (Table S1). In our study, cardiac anomalies increased the risk of death by 1.61 times (Table 2), but the result was not statistically significant in our study (p=0.168).
- Details on causes of death in the 52.5% of patients who died.
Response: The registered cause of death of our patients was the cardiocirculatory arrest caused by multiple organ failure. Only 38% of the patients underwent necroptic examination and the cause of death could be more accurately specified, but in the current study we did not analyze these aspects.
- There is no accepted definition of “giant omphalocele” and the authors use 5-7cm and severe viscero-abdominal disproportion to define this. How big were the omphaloceles of the patients in this study?
Response: In Table S1 (Supplementary materials) we added data about the size of the omphalocele, per decade and per total analyzed period. We also added explanations in the text (line 104-106 and 179-180).
- How long were patients followed in this study? With a 30-year cohort, what are long term outcomes? If these were recorded, would like to know about: Long-term feeding difficulties, Respiratory complications, Time to delayed closure (for the 41% managed conservatively), Rates of delayed mortality, and causes of death.
Response: As we answered above, we will soon have all the data available regarding the long-term evolution of our patients, and we intend to conduct another study that will answer these really important questions. Regarding the time to delay closure, usually we treat surgically the ventral hernia around the age of one year.
- As mentioned above, a table listing complication rates (including by decade) such as sepsis and renal failure would much better describe the population.
Response: From the total of 139 cases studied, 46 patients (33.09%) developed sepsis. From the total of 139 cases studied, 29 patients (20.86%) developed acute renal failure. We added these data in our manuscript -Table 1.
Minor Critiques
Methods
- The listed years of patient inclusion are 1991-2021, but Table 1 lists this as 1990-2020. This needs to be consistent.
Response: We corrected Table 1.
- Why is “associated abnormalities” defined as “two or more” abnormalities in addition to omphalocele? Is this what produced such a low incidence of “associated abnormalities” compared to the published literature?
Response: To statistically analyze the correlation between the presence of associated anomalies and death, we considered those patients who had at least two associated anomalies in addition to omphalocele (central nervous system, cardiac, renal, skeletal or genital abnormalities), as mentioned in the Methods chapter – line 95-96. In table 1, we added the rate of cardiac abnormalities and craniofacial dysmorphism.
- For patients who were managed by “conservative treatment,” why were they taken to the operating room for scarification treatment? Could this have been done at the bedside?
Response: Because our study covers a long period of time, in the first decades analyzed, our neonatal intensive care units did not have high-performance incubators and access inside the incubator was limited, making it difficult to handle the newborn for the correct disinfection of the omphalocele and the sterile dressing. That is why these maneuvers were performed in the operating room, at least until the patient was no longer dependent on the incubator. In the last decade, in fact, most of these patients were treated conservatively in the NICU – we made this clarification in our manuscript as well (line 110-113), thank you for your remark.
Results
- Should comment on the exclusion of 19 newborns with insufficient data in the results instead of in the methods
Response: We made the suggested modification.
- Hard to describe the rate of incidence as “sinusoidal” based on 3 data points – would probably remove this statement and just report the incidence in each decade
Response: We made the suggested modification.
- There are numerous paragraphs in the results explaining the statistical methods, as well as multiple tables that serve a similar purpose. Descriptions of statistical methods should be summarized more concisely and moved to the methods section. At least 2 of Tables 2-5 can be removed. This would allow for a demographics table (Table 1), outcomes table (Table 2), and 1-2 tables summarizing the logistic regression (Tables 3-4)
Response: We modified our manuscript and tables according to your recommendations; thank you for your suggestions. We only left 3 tables in the main text and added 4 tables in supplementary materials.
Discussion
- First or second paragraph of the discussion should summarize major results
Response: We added a paragraph on the beginning of the Discussion chapter (line 290-293).
- The authors comment on the increasing rate of prenatal detection over the course of the study, which I find interesting. I would like to know, in the discussion, about the extent of “routine” prenatal care in Romania, and how that is expanding, as it would help frame this evolution within the broader state of Romania’s healthcare system
Response: Dear reviewer, thank you for this interesting point you raised. We have added some phrases related to this aspect to the Discussions chapter (line 315-326) and three new references.
- The authors hypothesize that their higher mortality is related to lower rates of elective abortion, resulting in higher rates of associated anomalies (lines 311-314). However, the opposite is described in their data. This warrants more comment.
Response: It is well known from the specialized literature that genetic anomalies (such as trisomy 13, 18 or 1), major cardiac malformations and pulmonary hypoplasia associated with large omphalocele (conservative treatment) are risk factors for increased morbidity and mortality in case of omphalocele. We also demonstrated in our study that two or more associated abnormalities increase the mortality risk by 3.75 times (Table 2) and 6.16 times if the logistic regression is applied, cardiac malformations increase this risk 1.61 times (Table 2) and craniofacial dysmorphism 2.05 times (Table 2). If all these serious anomalies were detected antenatally and therapeutic abortion would be recommended, these newborns would no longer die postnatally, increasing neonatal mortality in our country. We added these explanations in the manuscript (line 335-351).
- The authors should address more limitations of their study in addition to its retrospective and single-center nature.
Response: We made the suggested addition.
- There are minor grammatical errors and results of awkward English translation throughout the manuscript
Response: We corrected the grammatical errors.
Thank you for reviewing our manuscript,
Elena Țarcă, MD, PhD
Round 2
Reviewer 1 Report
A few more suggestions;
Remove the word "period" from Table 1. Readers will find the table clear enough even without the mentioned word.
Instead of bold results, it might be better to mark a statistically significant result with "*" or "**"
In table 2, it is not necessary to repeat the word "death". “Df” does not need to be shown in the table. Write the abbreviations “OR” and "CI" in the table and an explanation below the table.
Also, in table 3 it is not necessary to specify "Df"
Author Response
Reviewer 1
A few more suggestions;
Remove the word "period" from Table 1. Readers will find the table clear enough even without the mentioned word.
Instead of bold results, it might be better to mark a statistically significant result with "*" or "**"
In table 2, it is not necessary to repeat the word "death". “Df” does not need to be shown in the table. Write the abbreviations “OR” and "CI" in the table and an explanation below the table.
Also, in table 3 it is not necessary to specify "Df"
Response
Dear Reviewer, thank you for your time and the positive remarks regarding our work.
We made all the suggested changes.
Reviewer 2 Report
I commend the authors of “Current challenges in the treatment of the omphalocele – Experience of a Tertiary Center from Romania” for making substantial revisions to their manuscript in response to their critiques. The authors have addressed many of my questions and concerns, including expanding their Chi-square analysis to account for other variables, listing the incidence of complications and management strategy by decade, and describing the evaluation of prenatal care in Romania over time.
I still have significant concerns:
· The authors did well to expand their Chi-square analysis to include other variables, some of which proved to be associated with death (hemorrhagic complications, blood transfusion, craniofacial dysmorphism). However, they did not then incorporate these variables into their regression model, and instead maintained their prior regression. This is contrary to what they described in their methods.
· Demographic information contributes to omphalocele outcomes in addition to complications such as sepsis and renal failure, e.g. prematurity would be expected to be a risk factor for mortality. These variables were apparently not included in these models.
· The authors note an association of conservative management with death. However, they also note in their methods that conservative management was pursued with larger defects, and in the setting of associated anomalies, both of which increase the risk of mortality. This implies that these variables are intrinsically associated, but this is not apparent or described in the results or discussion.
· I appreciate that the authors included cardiac abnormalities in their analysis and results. I was surprised to see that it did not correlate with death, since this is generally a widely-accepted risk factor. Much like sepsis and acute renal failure, “cardiac abnormality” should be defined in the methods, and perhaps a breakdown of abnormality type should be explored. For example, ASD or PDA may be included in “cardiac abnormalities” but have much less effect on outcome than more complex cardiac lesions such as tetralogy of Fallot or hypoplastic left heart syndrome.
· The authors continue to describe “associated abnormalities” as having 2 or more. Why is this? I would assume that having a chromosomal abnormality or complex cardiac anatomy alone would affect outcomes.
I would like to note that what the authors’ claim as unique about the study, that being tracking the evolution of care of omphalocele over time in Romania, is quite compelling. The majority of the data and its presentation do no emphasize this. Since this is the noted study strength, I think the authors could strategize as to how to highlight this with the robust data they have.

Author Response
Reviewer 2
Comments to Author
I commend the authors of “Current challenges in the treatment of the omphalocele – Experience of a Tertiary Center from Romania” for making substantial revisions to their manuscript in response to their critiques. The authors have addressed many of my questions and concerns, including expanding their Chi-square analysis to account for other variables, listing the incidence of complications and management strategy by decade, and describing the evaluation of prenatal care in Romania over time.
Response: Dear Reviewer, thank you for your time evaluating our manuscript and helped us improve our results.
I still have significant concerns:
- The authors did well to expand their Chi-square analysis to include other variables, some of which proved to be associated with death (hemorrhagic complications, blood transfusion, craniofacial dysmorphism). However, they did not then incorporate these variables into their regression model, and instead maintained their prior regression. This is contrary to what they described in their methods.
Response: Thanks to your suggestion, we redid the logistic regression model and tested all 8 potential predictors (statistically significant variables) presented in Table 2, finally obtaining a model with 6 independent variables (Sepsis, Acute renal failure, Associated abnormalities, Conservative treatment, Hemorrhagic disease and Prematurity).
- Demographic information contributes to omphalocele outcomes in addition to complications such as sepsis and renal failure, e.g. prematurity would be expected to be a risk factor for mortality. These variables were apparently not included in these models.
Response: We redid the logistic regression model and incorporated other variables, including prematurity.
- The authors note an association of conservative management with death. However, they also note in their methods that conservative management was pursued with larger defects, and in the setting of associated anomalies, both of which increase the risk of mortality. This implies that these variables are intrinsically associated, but this is not apparent or described in the results or discussion.
Response: We mentioned in the Methods chapter that we applied conservative treatment to patients with giant omphalocele or those with serious chromosomal abnormalities (trisomy 1, 13, 18) (line 110-113). Usually these severe chromosomal abnormalities are incompatible with life, and usually the omphalocele is small in these particular cases. Only the treatment is part of our final logistic regression model, and the model is tested for multicollinearity (line 174-179, 315-326 and Table S5).
- I appreciate that the authors included cardiac abnormalities in their analysis and results. I was surprised to see that it did not correlate with death, since this is generally a widely-accepted risk factor. Much like sepsis and acute renal failure, “cardiac abnormality” should be defined in the methods, and perhaps a breakdown of abnormality type should be explored. For example, ASD or PDA may be included in “cardiac abnormalities” but have much less effect on outcome than more complex cardiac lesions such as tetralogy of Fallot or hypoplastic left heart syndrome.
Response: We defined “cardiac abnormalities”, “hemorrhagic disease” and “prematurity” in our Methods section (line 98-102). Regarding cardiac abnormalities, 30 patients had minor abnormalities (including ASD, PDA) and only 29 had major cardiovascular malformations. We added this information in our manuscript (line 193-196).
- The authors continue to describe “associated abnormalities” as having 2 or more. Why is this? I would assume that having a chromosomal abnormality or complex cardiac anatomy alone would affect outcomes.
Response: Since among the associated anomalies we also included the minor ones (for example polydactyly or atrial septal defect), we decided that in the statistical analysis of correlation between the presence of associated anomalies and death, to consider those patients who had at least two associated anomalies. We mentioned that in the Methods chapter – line 96-97.
I would like to note that what the authors’ claim as unique about the study, that being tracking the evolution of care of omphalocele over time in Romania, is quite compelling. The majority of the data and its presentation do no emphasize this. Since this is the noted study strength, I think the authors could strategize as to how to highlight this with the robust data they have.
Response: We highlighted this aspect of our study at the beginning of the Discussions chapter (line 333-338) and in the section on the strength of the study (line 488-500).